# The Impacts of Aerosol Emissions on Historical Climate in UKESM1

Jeongbyn Seo [1], Sungbo Shim [1,*], Sang-Hoon Kwon [1], Kyung-On Boo [1], Yeon-Hee Kim [1], Fiona O'Connor [2], Ben Johnson [2], Mohit Dalvi [2], Gerd Folberth [2], Joao Teixeira [2], Jane Mulcahy [2], Catherine Hardacre [2], Steven Turnock [2], Stephanie Woodward [2], Luke Abraham [3,4], James Keeble [3,4], Paul Griffiths [3,4], Alex Archibald [3,4], Mark Richardson [5], Chris Dearden [5], Ken Carslaw [5], Jonny Williams [6], Guang Zeng [6] and Olaf Morgenstern [6]

[1] Innovative Meteorological Research Department, National Institute of Meteorological Sciences, Seogwipo 63568, Jeju, Korea; jbseo6822@gmail.com (J.S.); skysh2002@korea.kr (S.-H.K.); kyungon@korea.kr (K.-O.B.); yeonheekim@korea.kr (Y.-H.K.)

[2] Met Office Hadley Centre, Exeter EX1 3PB, UK; fiona.oconnor@metoffice.gov.uk (F.O.); ben.johnson@metoffice.gov.uk (B.J.); mohit.dalvi@metoffice.gov.uk (M.D.); gerd.folberth@metoffice.gov.uk (G.F.); joao.teixeira@metoffice.gov.uk (J.T.); jane.mulcahy@metoffice.gov.uk (J.M.); Catherine.Hardacre@metoffice.gov.uk (C.H.); steven.turnock@metoffice.gov.uk (S.T.); Stephanie.Woodward@metoffice.gov.uk (S.W.)

[3] National Centre for Atmospheric Science, University of Cambridge, Cambridge CB2 1EW, UK; nla27@cam.ac.uk (L.A.); jmk64@cam.ac.uk (J.K.); paul.griffiths@ncas.ac.uk (P.G.); Alex.Archibald@atm.ch.cam.ac.uk (A.A.)

[4] Department of Chemistry, University of Cambridge, Cambridge CB2 1EW, UK

[5] Centre for Environmental Modelling and Computation, University of Leeds, Leeds LS2 9JT, UK; M.G.Richardson@leeds.ac.uk (M.R.); earcdea@leeds.ac.uk (C.D.); K.S.Carslaw@leeds.ac.uk (K.C.)

[6] National Institute, for Water and Atmospheric Research, 6022 Wellington, New Zealand; jonny.williams@niwa.co.nz (J.W.); guang.zeng@niwa.co.nz (G.Z.); olaf.morgenstern@niwa.co.nz (O.M.)

[*] Correspondence: sbshim82@korea.kr

**Abstract:** As one of the main drivers for climate change, it is important to understand changes in anthropogenic aerosol emissions and evaluate the climate impact. Anthropogenic aerosols have affected global climate while exerting a much larger influence on regional climate by their short lifetime and heterogeneous spatial distribution. In this study, the effective radiative forcing (ERF), which has been accepted as a useful index for quantifying the effect of climate forcing, was evaluated to understand the effects of aerosol on regional climate over a historical period (1850–2014). Eastern United States (EUS), Western European Union (WEU), and Eastern Central China (ECC), are regions that predominantly emit anthropogenic aerosols and were analyzed using Coupled Model Intercomparison Project 6 (CMIP6) simulations implemented within the framework of the Aerosol Chemistry Model Intercomparison Project (AerChemMIP) in the UK's Earth System Model (UKESM1). In EUS and WEU, where industrialization occurred relatively earlier, the negative ERF seems to have been recovering in recent decades based on the decreasing trend of aerosol emissions. Conversely, the radiative cooling in ECC seems to be strengthened as aerosol emission continuously increases. These aerosol ERFs have been largely attributed to atmospheric rapid adjustments, driven mainly by aerosol-cloud interactions rather than direct effects of aerosol such as scattering and absorption.

**Keywords:** aerosol; effective radiative forcing; instantaneous radiative forcing; rapid adjustments; aerosol-radiation interaction; aerosol-cloud interaction

## 1. Introduction

Human activities, such as the emission of anthropogenic greenhouse gases (GHGs) and aerosols from industrial sources [1–3], have significantly affected the climate systems. The burning of fossil fuels has increased atmospheric carbon dioxide ($CO_2$) concentrations, rising from 280 ppm in the pre-industrial era to 410 ppm in the present day [4]. The combustion of fossil fuels also emits aerosols and their precursors, such as sulfur dioxides ($SO_2$), black carbon (BC), and organic carbon (OC). As a major contributing factor to human-induced climate change, GHGs and anthropogenic aerosols significantly perturb the radiative budget at the top of the atmosphere as well as at the surface, resulting in global temperature change. Particularly, the climate effects of human-induced aerosols have masked some of the warming induced by GHGs [5–7].

Aerosols affect the atmospheric radiation budget directly and indirectly. By the scattering and absorbing of solar radiation, aerosols can influence the shortwave radiation reaching the earth which is known as the direct effect or aerosol-radiation interaction. Aerosols can also modify the radiative budget through the aerosol-cloud interactions including aerosol indirect and semi-direct effects. The influences of the aerosol indirect effect are induced by the change in cloud properties such as cloud droplet number concentration (CDNC), cloud droplet effective radius (CDER). On the other hand, a semi-direct effect refers to cloud amount change as aerosol particles heat the atmosphere by absorbing solar radiation [8–10]. These radiative effects of aerosols depend on the properties of each chemical component. For example, sulfate aerosols mainly contribute to the radiative cooling effect by scattering incoming solar radiation, while BC has an atmospheric heating effect by absorbing the radiation [11–13]. They also interact with clouds in different way. The aerosol-cloud interactions of sulfate aerosols are mainly driven by the change in cloud microphysics, but BC interacts with cloud through perturbing atmospheric temperature profile. Human-induced aerosols affect not only climate change, but a wide range of sectors such as air quality, atmospheric visibility, and human health around the world [14,15]. Because of widespread impacts on human society, many countries and communities have made considerable efforts to reduce aerosol pollution [16,17].

Eastern United States (EUS) and Western European Union (WEU) achieved rapid industrial development during the mid-20th century [18,19]. In this period, aerosols played a key role in the global climate systems due to their radiative cooling effect with a large uncertainty ranging from −0.22 to −1.85 W/m$^2$ [3,8,9,20]. Since the 1980s, aerosol concentrations in EUS and WEU have been declining, mainly due to reduced local aerosol and precursor emissions. In contrast, Eastern Central China (ECC), with rapid development in recent decades, has become a major source of anthropogenic aerosols, exceeding the current emission levels of EUS and WEU [21,22]. The different stages of development have mainly resulted in different timings and magnitudes of aerosol effects, which have been crucial in shaping the climate change impacts on specific sectors of regional interests.

To quantify the effects of aerosols on climate, the stratosphere-adjusted definitions of radiative forcing (RF) have been used for a long time [23,24]. As an updated definition, the effective radiative forcing (ERF), including all tropospheric and land-surface adjustments, and stratospheric temperature adjustments, has been introduced in the fifth report of the Intergovernmental Panel on Climate Change. Accepted as a more effective indicator than RF for estimating potential climate response to imposed forcing, the ERF has been used in many studies to quantitatively assess the impact of forcing on climate [15,25–27]. Unlike most previous studies, the ERF uses the fixed-SST (Sea Surface Temperature) method, in which SSTs are held fixed at pre-industrial levels under equilibrium states. This study attempts to understand the ERF using the historical SST method to consider more realistic variations in aerosol forcing from transient simulations. In addition, the use of historical SSTs rather than pre-industrial SSTs can eliminate any effects of using an inconsistent background climate state, such as different cloud cover and natural emissions [28].

The aim of this study is to understand the effects of aerosols on climate over a historical period (1850–2014) using the UK's Earth System Model (UKESM1). By diagnosing transient ERFs, both globally, and in each study region (EUS, WEU, ECC), we attempt to quantify the global/regional climate impacts

of aerosol and understand the changing trend over time. Additionally, we analyze spatial patterns of forcing and responses to understand how the Earth's system responds to regional differences in aerosol forcing and disentangles the main contributor of the change. Section 2 describes the model and data used in this study. The main results, including the transient ERF due to anthropogenic aerosols in the historical period, are presented in Section 3, while Section 4 summarizes the study.

## 2. Experiments

### 2.1. Model

The UK's Earth System Model (UKESM1) that participated in Coupled Model Intercomparison Project 6 (CMIP6) is used in this study. UKESM1 is comprised of the Global Atmosphere 7.1/Global Land 7.0 (GA7.1/GL7.0) configuration of the Hadley Centre Global Environment Model version 3 (HadGEM3) [29,30]. The Earth System model also encompasses marine and terrestrial biogeochemical cycles and fully interactive stratosphere–troposphere chemistry [31] from the UK Chemistry and Aerosol (UKCA) model [32,33]. The GLOMAP-mode [34] is employed as a microphysical aerosol scheme. The model can simulate the whole-atmosphere chemistry and the indirect effects of aerosol on climate, such as aerosol-cloud interactions [31,34,35]. The horizontal resolution is N96 (~135 km) and the model has 85 vertical levels on a terrain-following hybrid height coordinate. More detailed descriptions of the model are included in Seller et al. (2019) [36].

Recent studies have conducted multi-model comparison using CMIP6 models. As compared multi-model ensemble of aerosol ERF ($-1.04$ W/m$^2$), Smith et al. (2020) [37] have shown that UKESM1 have been placed in the range of CMIP6 with relatively strong sensitivity to aerosol forcing ($-1.21$ W/m$^2$). Although UKESM1 seems to be have a slightly higher BC ERF than other CMIP6 models, all aerosol ERFs are still in the range of the multi-model ensemble [38].

### 2.2. Experiments

In this study, we used GCM (Global Climate Model) simulations implemented within the framework of the Aerosol and Chemistry Model Intercomparison Project (AerChemMIP) [28], endorsed by the Coupled Model Intercomparison Project 6 (CMIP6) [39]. The AerChemMIP is designed to quantify the climate and air quality impacts of aerosols and chemically reactive gases to facilitate the understanding of the contribution of aerosols and chemistry to climate change. All simulations used in this study are listed in Table 1. The emissions and GHG concentrations for 1850 and 2014 were followed by Hoesly et al. (2018), van Marle et al. (2017), and Meinhausen et al. (2017) [40–42]. The analysis has been conducted using monthly data for global, EUS (97–70° W, 28–50° N), WEU (13° W–15° E, 35–57° N), and ECC (101–130° E, 19–41° N) domains.

To estimate the time evolution of ERF ($ERF_{trans}$) over the historical period (1850–2014), the transient simulations newly introduced in CMIP6 were analyzed. These simulations have been designed with prescribed historical SSTs and sea ice fields to calculate the ERFs due to aerosol forcing: the histSST (with all forcings as historical) and the histSST-piAer (with all forcings as historical but using aerosol precursor emissions of the year 1850). The monthly mean time-evolving SSTs and sea ice from the historical simulations have been prescribed. The simulations covered a historical period of 165 years (1850–2014) and the emission of major aerosol precursor gases (SO$_2$) and primary aerosols (BC, OC) over this period are shown in Figure 1. The radiative effects of the aerosol emissions were estimated by subtracting the histSST-piAer simulations from the histSST simulations.

**Table 1.** List of all the fixed historical sea surface temperature effective radiative forcing (ERF) experiments used in the study to diagnose the pre-industrial to historical period ERFs from changes in aerosol emissions.

| Experiment ID | MIP | $N_2O$ | $CH_4$ | Aerosol Precursors | Trop. $O_3$ Precursors |
|---|---|---|---|---|---|
| histSST | AerChemMIP | Hist | Hist | Hist | Hist |
| histSST-piAer | AerChemMIP | Hist | Hist | 1850 | Hist |
| piClim-control | CMIP6 | 1850 | 1850 | 1850 | 1850 |
| piClim-Aer | AerChemMIP | 1850 | 1850 | 2014 | 1850 |
| piClim-$SO_2$ | AerChemMIP | 1850 | 1850 | 1850 (non-$SO_2$) 2014 ($SO_2$) | 1850 |
| piClim-BC | AerChemMIP | 1850 | 1850 | 1850 (non-BC) 2014 (BC) | 1850 |
| piClim-OC | AerChemMIP | 1850 | 1850 | 1850 (non-OC) 2014 (OC) | 1850 |
| sstClim | CMIP5 | 1850 | 1850 | 1850 | 1850 |
| sstClimAerosol | CMIP5 | 1850 | 1850 | 2000 | 1850 |

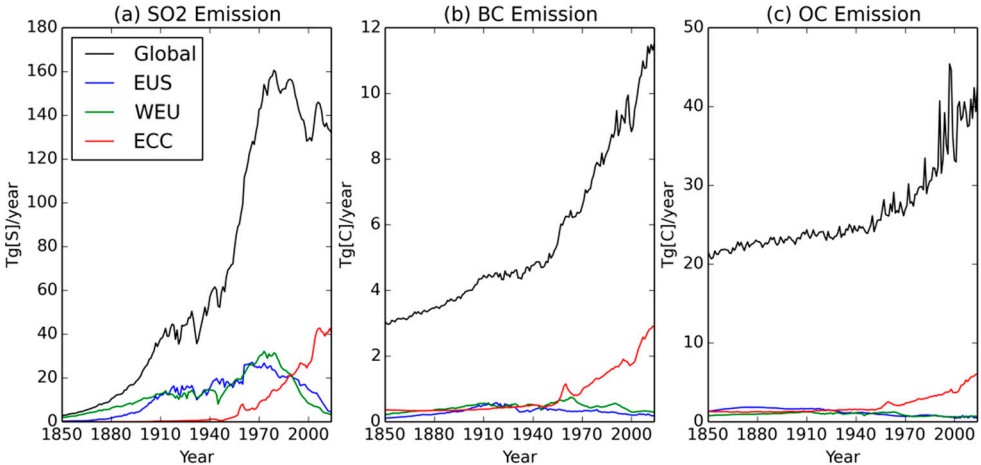

**Figure 1.** Timeseries of global and regional (**a**) $SO_2$, (**b**) BC, and (**c**) OC emissions over the historical period (1850–2014).

To compare the transient ERF with the conventionally defined present-day ERF ($ERF_{fsst}$) following the recommendations from Forster et al. (2016) [27], two other sets of timeslice experiments which prescribed a monthly averaged climatology of SST and sea ice from pre-industrial simulations (piClim-control, piClim-Aer) were also used. The primary aerosol and aerosol precursor emissions of the piClim-control and piClim-Aer simulations were held constant at pre-industrial (1850) and present-day (2014) levels. Additional perturbation experiments (piClim-$SO_2$, piClim-BC, piClim-OC) parallel to piClim-control, in which the emission of specific aerosol types ($SO_2$, BC, OC) was changed from 1850 to 2014, were used to understand the difference in regional response. All these timeslice simulations covered 45 years in length and the latter 30 years of the simulations were diagnosed. The $ERF_{fsst}$ value of the year 2000 from the ensemble of CMIP5 5 models (CanESM2, CSIRO-Mk3-6-0, GFDL-CM3, IPSL-CM5A-LR, NorESM1-M) was also compared. We analyzed the timeslice simulation pairs prescribing climatological SSTs and sea ice imposed from pre-industrial simulations with aerosol emissions at 1850 (sstClim) and 2000 (sstClim-Aerosol).

*2.3. ERF*

The ERF has been defined as the difference (Δ) in the net TOA (top of atmosphere) radiative flux (F) between the perturbed simulation and the control simulation. It can be separated into a component

due to the aerosol instantaneous radiative forcing (IRF), any non-aerosol changes in clear-sky flux ($ERF_{cs,clean}$), and cloud property changes ($\Delta CRE'$). To distinguish the contribution of each process to establish the ERF, we focused on the IRF, driven mainly by aerosol-radiation interactions and rapid adjustment (RA), including atmospheric adjusted responses. These adjustments include α change in stratospheric temperature as well as adjustments such as tropospheric temperature, water vapor, clouds, and land surface albedo. Each term has been calculated as recommended in Ghan et al. (2013) [43]:

$$
\begin{aligned}
ERF &= \Delta(F - F_{clean}) + \Delta F_{clear,clean} + \Delta\left(F_{clean} - F_{clear,clean}\right) \\
&= IRF + ERF_{cs,clean} + \Delta CRE' \\
&= IRF + RA
\end{aligned}
\tag{1}
$$

The "clean" indicates the fluxes calculated by the double-call radiation code which excludes aerosol-radiation interactions from any aerosols.

## 3. Results

### 3.1. Aerosol Optical Depth

Figure 2a shows the global and regional time series of annual-mean change in aerosol optical depth (AOD) over the historical period (1850–2014) relative to the pre-industrial (PI) (1850) level. It implies that there are regional differences in the main timing and intensity of aerosol emissions. In EUS and WEU, where industrialization occurred in a relatively early period, the AOD began to increase from the early 20th century. The increasing trend continued to the mid-20th century and the largest AOD is shown in 1970. On the other hand, ECC, which had lower emissions in the early 20th century, shows a dramatic increase after 1950. From the late 20th century, the AOD in EUS and WEU shows a decreasing trend due to strong air quality policies, while it continuously increases in ECC. These regional contrasts have weakened the increasing trends of global-mean AOD in the late 20th century.

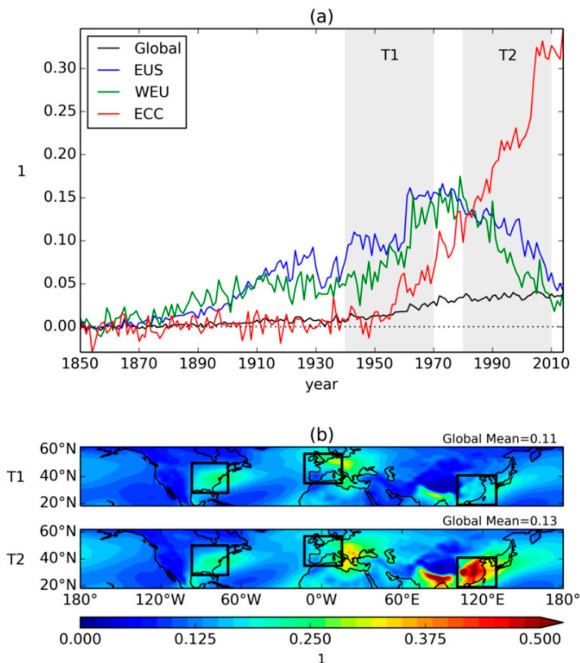

**Figure 2.** (**a**) Timeseries of global and regional mean change in aerosol optical depth (δAOD) at 550 nm over the historical period (1850–2014) relative to pre-industrial (PI) (1850) levels with grey shade indicating analysis periods T1 (1940–1970) and T2 (1980–2010), and (**b**) the spatial distribution of the 550 nm AOD. Each box indicates Eastern United States (EUS), Western European Union (WEU), and Eastern Central China (ECC) region.

In this study, the T1 period (1940–1970), when all the major regions emitting anthropogenic aerosols have an increasing AOD, and the T2 period (1980–2010), when AOD decreases in EUS and WEU and continuously increases in ECC, are analyzed separately. Figure 2b shows the spatial distribution of AOD in the T1 and T2 periods. Despite the reversed changing trend between T1 and T2 periods, a similar AOD on average is shown in EUS and WEU. The largest AOD in the world is shown in ECC over the T2 period as the aerosols increase dramatically.

*3.2. ERFs*

The transient responses of global and regional ERFs due to aerosol forcing over the historical period have been analyzed (Figure 3). The ERF is further decomposed into IRF and RA which are estimated by Equation (1). The IRF includes aerosol-radiation interactions (i.e., scattering and absorption) and the RA is driven mainly by aerosol-cloud interactions. As the AOD increases in all the domains (Figure 2a), regional aerosol ERF shows a decreasing trend, exerting a global ERF of approximately $-1.03$ W/m$^2$ in the T1 period. The regional effect of aerosol changes in this period is much greater than the global effect. The decreasing trend of aerosol ERFs have been shown in all the regions. Particularly, WEU ($-0.10$ W/m$^2$/yr) show a stronger aerosol ERF trend than EUS ($-0.08$ W/m$^2$/yr) and ECC ($-0.07$ W/m$^2$/yr). In the T2 period, the negative aerosol ERFs have been rapidly recovered in EUS and WEU as regional AOD decreased. Since they still have a negative ERF compared to the PI period, the radiative cooling effect of aerosols has been prominent even in the recovery state. In contrast, the decreasing trend of aerosol ERF in ECC ($-0.10$ W/m$^2$/yr) becomes stronger as the AOD continuously increases. As these regional responses cancel each other out, global-mean aerosol ERF ($0.01$ W/m$^2$/yr) seems to be relatively constant in the late 20th century.

The $ERF_{fsst}$ for 2000/2014 calculated from the equilibrium run in CMIP5/CMIP6 is also denoted in Figure 3 (cyan and green circles, respectively). Compared with the $ERF_{trans}$ for 2000/2014 calculated from transient simulations, the values are similar each other. This means that the value for ERF can also be reasonably estimated by transient simulations as with conventional methods using equilibrium runs. Estimation methods that use transient simulations have the additional advantage of showing the evolution of ERF changes over time.

Global and regional ERFs are more dominated by the atmospheric RA term (red) than IRF (blue), including the direct aerosol effect. Approximately 60% of aerosol ERFs come from the RA, driven mainly by aerosol-cloud interactions (Table 2). This is consistent with the previous studies [37,44], showing that the radiative cooling effect of aerosol-cloud interactions is more dominant than that of aerosol-radiation interactions. In particular, the IRF in ECC is much smaller than other regions, even when the AOD is largest in the late 20th century.

**Table 2.** Global and regional means of effective radiative forcing (ERF, W/m$^2$), instantaneous radiative forcing (IRF, W/m$^2$), and rapid adjustment (RA, W/m$^2$), including a component due to any non-aerosol change in clear-sky flux ($ERF_{cs,clean}$, W/m$^2$) and cloud property changes ($\Delta CRE'$, W/m$^2$) for the T1 (1940–1970) and T2 (1980–2010) periods relative to the pre-industrial period (PI) (1850), including an estimate of the standard error (Forster et al., 2016).

| | RF Relative to PI | | Glob | EUS | WEU | ECC |
|---|---|---|---|---|---|---|
| **T1** | | ERF | $-1.03 \pm 0.05$ | $-5.04 \pm 0.27$ | $-2.89 \pm 0.37$ | $-2.34 \pm 0.34$ |
| | | IRF | $-0.16 \pm 0.01$ | $-1.44 \pm 0.09$ | $-0.92 \pm 0.08$ | $0.10 \pm 0.04$ |
| | RA | $ERF_{cs,clean}$ | $0.00 \pm 0.02$ | $-0.46 \pm 0.19$ | $0.37 \pm 0.15$ | $0.08 \pm 0.22$ |
| | | $\Delta CRE'$ | $-0.87 \pm 0.04$ | $-3.14 \pm 0.27$ | $-2.34 \pm 0.40$ | $-2.52 \pm 0.41$ |
| **T2** | | ERF | $-1.43 \pm 0.05$ | $-4.57 \pm 0.32$ | $-3.23 \pm 0.38$ | $-3.87 \pm 0.32$ |
| | | IRF | $-0.30 \pm 0.01$ | $-1.63 \pm 0.06$ | $-1.23 \pm 0.09$ | $-0.74 \pm 0.08$ |
| | RA | $ERF_{cs,clean}$ | $0.04 \pm 0.03$ | $-0.00 \pm 0.23$ | $0.45 \pm 0.17$ | $0.12 \pm 0.18$ |
| | | $\Delta CRE'$ | $-1.17 \pm 0.03$ | $-2.94 \pm 0.30$ | $-2.45 \pm 0.40$ | $-3.24 \pm 0.32$ |

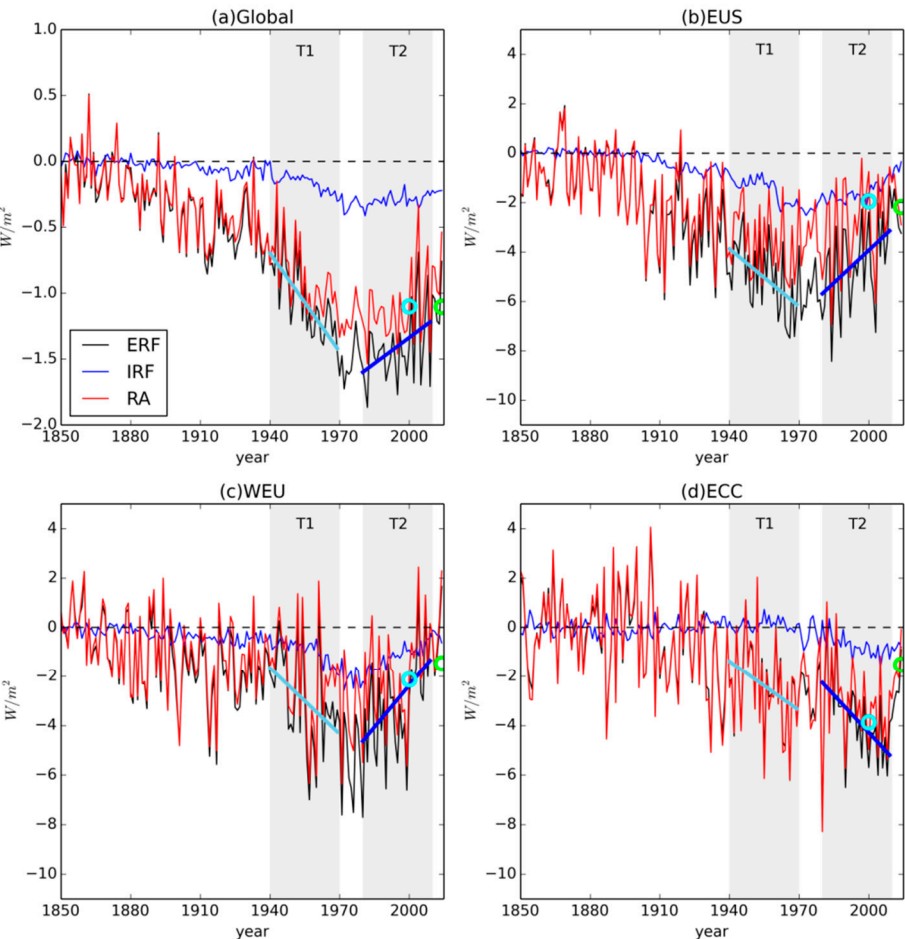

**Figure 3.** Global (**a**) and regional (**b**–**d**) means of effective radiative forcing (ERF), instantaneous radiative forcing (IRF), and rapid adjustment (RA) over the historical period (1850–2014) estimated by transient simulations ($ERF_{trans}$) with grey shade indicating analysis periods T1 (1940–1970) and T2 (1980–2010). Solid lines indicate the trends of ERF in T1 (1940–1970) and T2 (1980–2010) periods with cyan and green colored circles indicating ERF for 2010 in CMIP5 and 2014 in the Coupled Model Intercomparison Project 6 (CMIP6) calculated from the equilibrium run ($ERF_{fsst}$).

The significant contribution of the RA to ERF is also shown in the time-mean value for the T1 and T2 periods (Table 2). The aerosol IRF and RA are negative in most regions as AOD increases higher than during PI levels. The IRF, mainly driven by aerosol-radiation interactions, contributes to approximately 30% of the aerosol ERFs in EUS and WEU, irrespective of the period. On the other hand, the aerosol IRF over the T1 period has a radiative warming effect (+0.10 W/m²), which is opposite to the negative aerosol ERF in ECC, despite increase in AOD. In ECC, the aerosol IRF over the T2 period also has a small contribution (approximately 19%) to the ERF relative to other regions. The relatively lower contributions of aerosol IRF to ERF in ECC, even though there is larger AOD there than in other regions, seems to be contradictory. To understand these regional differences, the IRFs for each aerosol species (SO₂, BC, OC, and Aer) have been analyzed.

Figure 4 shows the IRFs for individual aerosol species (SO₂, BC, and OC) and total aerosols (Aer). As the analysis has been conducted using timeslice simulations with present-day levels (2014), the magnitude of responses is closely related with the regional emissions shown in Figure 1. The strongest responses are shown in ECC where the highest amounts of aerosols are emitted, while EUS and WEU show relatively weak responses.

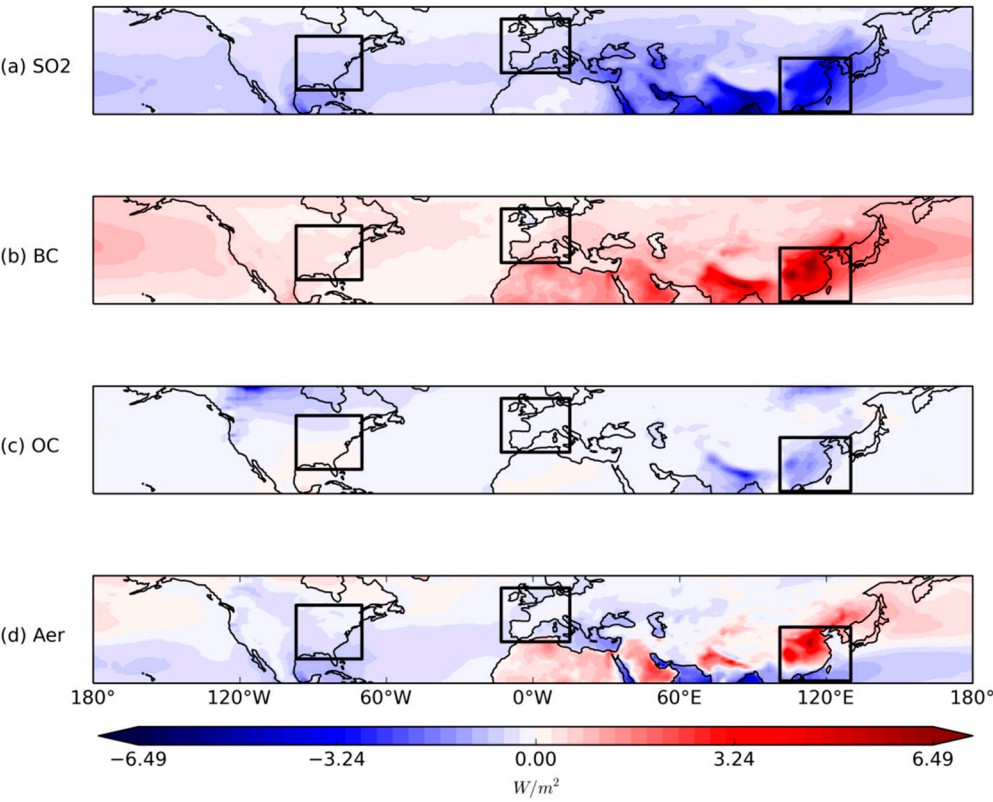

**Figure 4.** Geographical distribution of the present-day (PD) (2014) instantaneous radiative forcing (IRF) relative to the pre-industrial period (PI) (1850) for (**a**) $SO_2$, (**b**) BC, (**c**) OC, and (**d**) all aerosols in $W/m^2$.

In the case of $SO_2$ and OC, aerosols create the radiative cooling effect by reflecting incoming solar radiation while BC produces atmospheric warming by absorption of the radiation. As the $SO_2$ and BC seems to be the main contributors of aerosol IRF, the responses of these two species have been compared with each other. The aerosol IRF in EUS and WEU (−0.38, −0.23 $W/m^2$) becomes negative due to the larger radiative cooling effect of $SO_2$ (−0.63, −0.48 $W/m^2$) than the BC warming effect (+0.31, +0.34 $W/m^2$). Contrarily, the positive aerosol IRF (+0.74 $W/m^2$), shown in ECC seems to be due to a stronger absorbing effect of the BC (+2.98 $W/m^2$) than the $SO_2$ cooling effect (−2.55 $W/m^2$). Thus, the main reason for less effective IRF in ECC is that the cooling effect of scattering aerosols is almost offset by BC warming, as shown by O'Connor et al. (2020) [45].

Lastly, the spatial distribution of aerosol ERFs and the main contributing factors were investigated. Only figures for the recent period (T2) are shown here, but the results of the T1 period showing an opposite changing pattern in EUS and WEU are also consistent. The aerosol ERFs are broken down into the IRF driven by aerosol direct effect and the RA in this study. As the change in shortwave radiation consists of over 80% of the IRF and RA, the main contributors of $IRF_{sw}$ and $RA_{sw}$ have been analyzed.

The spatial distributions of the shortwave component in the IRF ($IRF_{sw}$) and AOD change are shown in Figure 5. In EUS and WEU, the increase in AOD seems to be highly correlated with the reduction in $IRF_{sw}$. This significant negative correlation indicates that the main contributor of $IRF_{sw}$ is the reduction in radiation due to the increased scattering of aerosols which reflect incoming shortwave radiation. On the other hand, a lower spatial correlation is shown in ECC where the effect of absorbing aerosols is larger than other regions. The negative correlation between AOD and $IRF_{sw}$ as in other regions is shown in ocean areas, while the opposite pattern is shown inland due to the dominant effect of absorbing aerosols.

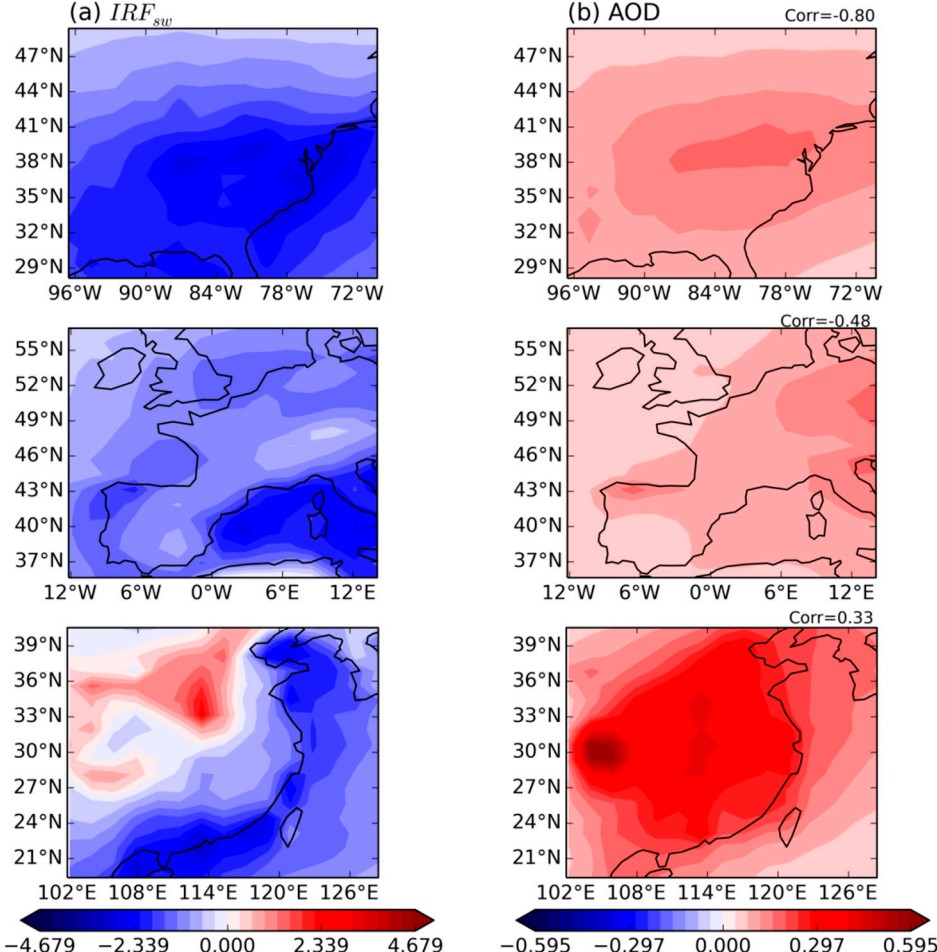

**Figure 5.** Regional (1st: EUS, 2nd: WEU, and 3rd: ECC) distributions of (**a**) the shortwave instantaneous radiative forcing (IRFsw, W/m$^2$) and (**b**) change in aerosol optical depth (AOD, 1) at 550 nm from the pre-industrial (1850) to the T2 period (1980–2010). Spatial correlation coefficients are shown in the upper right corner.

As shown in Figure 3, the RA—mainly composed of shortwave components ($RA_{sw}$)—contributes largely to the aerosol ERFs. The decomposition of the RA into a component due to any non-aerosol change in clear-sky flux ($ERF_{cs,clean}$) and cloud property changes ($\Delta CRE'$) shows that the aerosol-cloud interactions are the major constituent process to the RA (Table 2). Therefore, the change in total clouds according to ISCCP (International Satellite Cloud Climatology Project) classification has been analyzed here as one of the main contributors to $RA_{sw}$ (Figure 6). In EUS and WEU, the negative $RA_{sw}$ seems to be highly correlated with the increase in total clouds. Consistent with the previous studies showing that the radiative cooling due to the aerosol-cloud interactions is the main factor to determine the aerosol ERFs [40,41], the negative $RA_{sw}$ seems to be closely related with the increase in clouds, and especially stratus clouds ($p_{top} < 680$ hPa and $\tau > 23$, Figure 7). The increase in stratus clouds can be explained by the stabilization of the atmosphere due to surface cooling by aerosols [46].

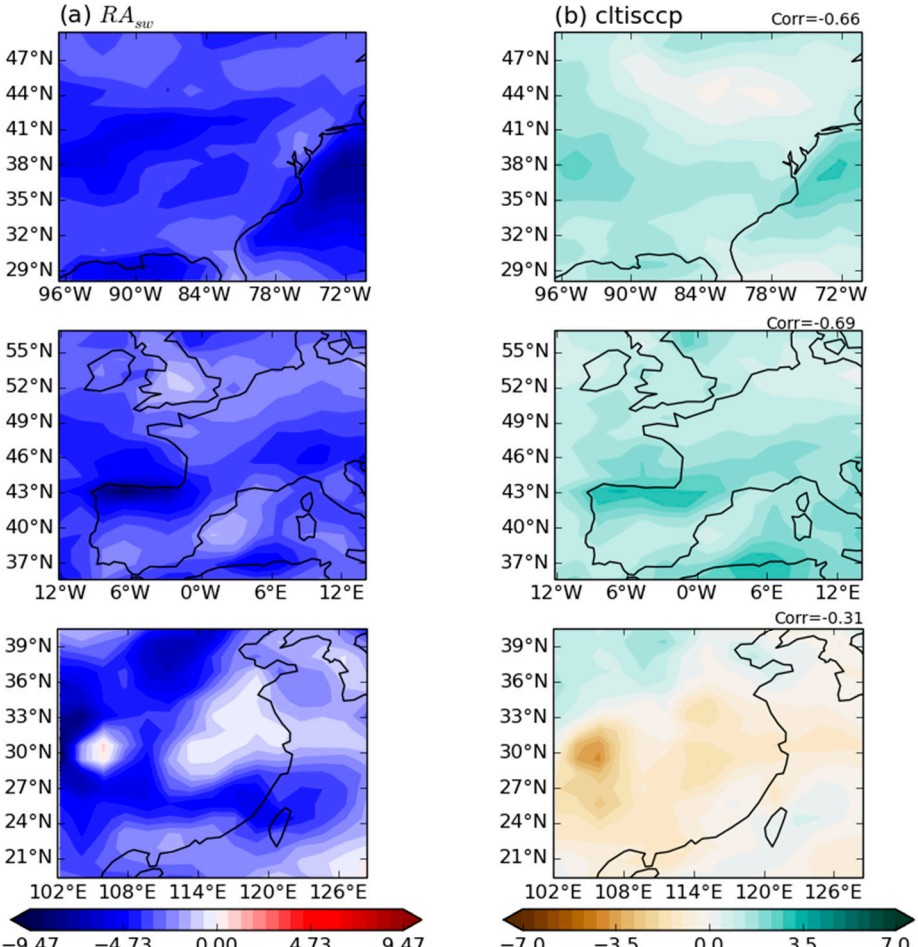

**Figure 6.** Regional (1st: EUS, 2nd: WEU, and 3rd: ECC) distributions of (**a**) the shortwave rapid adjustment (RAsw, W/m$^2$) and (**b**) the change in ISCCP total cloud fraction (cltisccp, %) from the pre-industrial (1850) to the T2 (1980–2010) period. Spatial correlation coefficients are shown in the upper right corner.

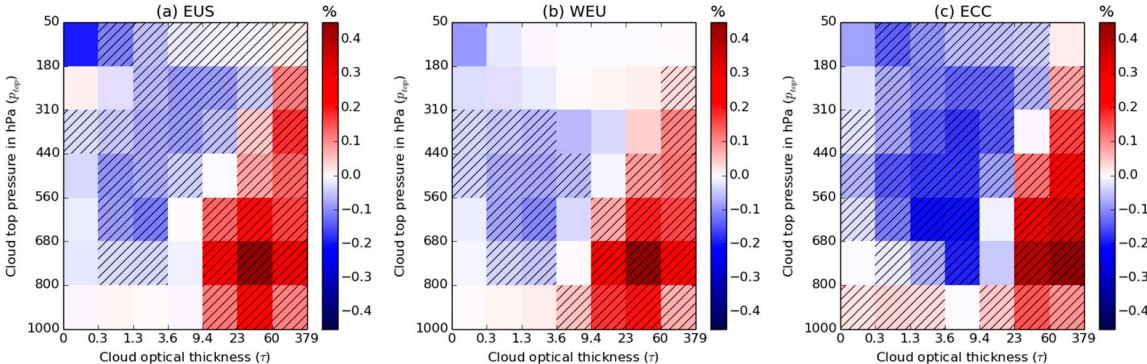

**Figure 7.** Regional (1st: EUS, 2nd: WEU, and 3rd: ECC) mean change in ISCCP-simulated cloud fraction in CTP (Cloud Top Pressure) -τ space for the T2 period (1980–2010) relative to the pre-industrial period (1850). The 90% significance denoted as dashed lines.

As the stratus cloud is known for its radiative cooling effect by reflecting incoming solar radiation, these increases in clouds contribute to the negative $RA_{sw}$ [47,48]. In ECC, the $RA_{sw}$ is also negative as in other regions while there is a lower spatial correlation with change in total cloud. Unlike other regions where the clouds generally increase, ECC shows the reduction in the clouds on the southern

side of 33° N. To understand these regional differences, the simulated ISCCP cloud fraction changes in the model have been analyzed (Figure 7). The increase in stratus cloud is shown as in other regions, but there is a difference in that the strong decrease in the middle cloud (440 hPa $< p_{top} <$ 680 hPa) simultaneously occurs in ECC. The increase in stratus clouds is particularly dominant on the northern side of 33° N, while both the increase in stratus clouds and the reduction in middle clouds are strongly shown on the southern side (figures not shown). This indicates that the increased stratus cloud contributes to negative $RA_{sw}$ as in other regions, although the total cloud amount seems to be decreased due to the large reduction in the middle clouds on the southern side of ECC.

## 4. Conclusions

As anthropogenic aerosols have contributed to large uncertainties in future climate prediction, it is important to understand and quantify the aerosol effects on climate. ERF is one important measure of the climatic effects of aerosols. In this study, we have analyzed how aerosol emissions change and quantified the effect since the PI period with the UK's Earth System Model (ESM), UKESM1, using the concept of ERF.

Aerosols exert a cooling effect with a negative ERF in the early to mid-20th century. In EUS and WEU, where industrialization occurred relatively earlier, the negative ERF seems to be recovering in the late 20th century as aerosol emissions decrease due to air quality controls. On the other hand, the radiative cooling in ECC seems to be strengthened as the aerosol emissions continuously increase. These contrasting patterns in each region make the global annual-mean ERF relatively constant in recent decades.

The aerosol ERFs have been broken down further into IRF and RAs to quantify their role on the composition of the ERF. While approximately 30% of the aerosol ERF in other regions consists of IRF by aerosol-radiation effects, relatively small contributions of IRF have shown in ECC despite the largest aerosol emissions. This seems to be driven by the larger effect of absorbing aerosols in ECC unlike other regions. Regardless of the regions, RA mainly from the aerosol-cloud interactions have been shown to be a dominant contributor to the aerosol cooling effects. The aerosol-cloud interactions include the response of the change in cloud amount, cloud albedo, etc. The effects of the cloud amount change have been analyzed in this study. As the atmosphere become stable due to surface cooling by aerosols, the amount of stratus cloud increases. These increases in cloud amounts have contributed significantly to the radiative cooling effect (negative ERF) by reflecting incoming solar radiation.

The results are consistent with another study [49]. Using the difference in equilibrium simulations that prescribed aerosol emissions in 1970 and 2010, they show different changing trends in aerosol ERFs. In Europe and US regions, the cooling effect of aerosols seems to have recovered in recent decades while the effect becomes stronger in Asia. The main contributors of aerosol ERFs have been analyzed as the aerosol-cloud interactions, also consistent with our study. Compared to these similar studies, this study is additionally meaningful in that the time evolutions of aerosol ERFs over the historical period have been analyzed using experiments newly introduced to CMIP6. While the cooling effect of aerosols has been recovering in EUS and WEU over recent decades due to robust air quality policies, strong radiative cooling by direct and indirect aerosol effects has been shown in ECC. As the aerosol emissions in East Asia will also be expected to decrease in the future due to air quality controls, the radiative cooling effect of aerosols will be recovered, inducing accelerated global warming. The analysis for the future prediction according to the aerosol emission changes will be conducted in future study. This study also emphasizes the importance of rapidly adjusted atmospheric process by aerosols. However, there are limitations with the single model used. The models participating in CMIP6 will be analyzed in future studies to improve reliability. Additionally, additional components in RAs such as the response of the change in water vapor and cloud albedo need to be considered further for comprehensive understanding of the effects of aerosols.

**Author Contributions:** The model simulations were ran by J.S., S.S., S.-H.K., F.O., B.J., M.D., G.F., J.T., J.M., C.H., S.T., S.W., L.A., J.K., P.G., A.A., M.R., C.D., K.C., J.W., G.Z. and O.M. The manuscript and graphics were prepared by J.S., S.S. and S.-H.K. The discussion and revision of the manuscript were contributed by S.S., S.-H.K., K.-O.B. and Y.-H.K. with additional contributions from all co-authors. All authors have read and agreed to the published version of the manuscript.

**Funding:** This work was funded by the Korea Meteorological Administration Research and Development Program "Development and Assessment of IPCC AR6 Climate Change Scenario" under Grant (KMA2018-00321).

**Conflicts of Interest:** The authors declare no conflict of interest.

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
