# Peer review of "The Impacts of Aerosol Emissions on Historical Climate in UKESM1"

_atmosphere, doi:10.3390/atmos11101095_

Round 1
Reviewer 1 Report
The authors have addressed my comments adequately.
Reviewer 2 Report
Thank you for your account of how UKESM1 compares to other CMIP6 models. However, I meant that this information should be included in the manuscript for the sake of the readers - not just for my personal curiosity as a reviewer. This could be easily included in Section 2.1.
Other than that I am happy with how the authors have addressed my comments.
Author Response
Please see the attachment.

This manuscript is a resubmission of an earlier submission. The following is a list of the peer review reports and author responses from that submission.
Round 1
Reviewer 1 Report
This paper analyzed the monthly mean data of UKESM1 simulations to study the impacts of aerosol emissions on historical climate. The effect of aerosols over different regions has been discussed in many studies. This study has made some progress on the model data analysis and perhaps contribute to the field. However, the new idea and finding from this study should be emphasized since this subject is quite popular. Overall, this manuscript is not recommended for publication in Atmosphere.

Reviewer 2 Report
The manuscript presents aerosol changes and their associated climate effects in the recent decades in the UK ESM climate model. The model was recently developed, so the results in this study provide some new assessment of aerosol effects on climate. The paper is clearly written and easy to follow. I will recommend its publication on the journal Atmosphere, after the following comments can be addressed.
- Several similar studies have been done using different climate models, so the present study needs to put their assessment into full comparison with those existing literature. For example, Wang et al. (2015, “Atmospheric Responses to the Redistribution of Anthropogenic Aerosols”) reported aerosol ERF changes since the 1970s over different regions using CESM1. They also compared the simulated aerosol-induced surface temperature trends with observed ones over Europe, US, China, and India, and broke down ERF to direct and indirect forcing. Some comparative discussions are needed.
- Can the authors present some results about precipitation changes under different aerosol forcing scenarios from UKESM1?
- It is somewhat surprising to see that the net IRF over China is positive in this model, which means BC absorption is quite strong. This makes me wonder who BC mixing state are treated in the model? Recent climate modeling studies clearly show the strong dependence of BC forcing on the mixing state assumption and aging process parameterizations (Wang et al., 2018 “Constraining Aging Processes of Black Carbon in the Community Atmosphere Model Using Environmental Chamber Measurements”).
- Cloud changes in Fig. 7 are quite small in magnitude. How significant are they in statistics?
Reviewer 3 Report
The manuscript submitted by Seo et al. investigates the regional climate effect of aerosols in the period of 1850 to 2014 using AerChemMIP simulations by the UKESM1 model. They find links between trends in anthropogenic aerosol emissions and the ERF, with different timings in different regions, consistent with e.g. the European air pollution clean-up as well as with the continued emission increase in parts of Asia. They find that changes in rapid adjustments, rather than aerosol-radiation interactions, is what dominate the ERF variations. While these are not groundbreaking results, they make a fine addition to earlier findings on the topic. It is good to see the interesting AerChemMIP simulations in use. I have added a list of comments below.
Specific comments:
- L 39: “climate systems” –> “the climate system”
- L 40: please add “ppm” behind the number 280
- L 44: “radiative budget” --> “the radiative budget”
- L 49-50: While I agree that scattering aerosols will reduce the amount of solar radiation absorbed by the future, and that absorbing aerosols heat the atmosphere, the absorbing aerosols also (at least initially) cool the surface as they hinder the radiation to reach the surface by absorbing it further up. A subtle rewording would be good.
- L 51: Aerosol-cloud interactions include several processes, including the one often referred to as the semidirect effect. The text “aerosol-cloud interactions, know as aerosol indirect radiative forcing”, is therefore not correct. It would be good to instead list the processes we typically view as aerosol-cloud interactions, and perhaps also comment upon how sulfate versus BC influence these. For instance, while sulfate tend to interact strongly with cloud microphysics and thus is important to the aerosol indirect radiative forcing, BC has a stronger influence on clouds through its effect on the atmospheric temperature profile.
- L 57-58: “played key roles in climate systems” --> “played a key role in the global climate system”
- L 66: The first part of the sentence is a bit off – it sounds like the disturbing helps you quantify the effect. I suggest just deleting “by disturbing the earth’s radiation budget” as the sentence makes sense without it.
- L 98: As this is a single-model study, I feel that we lack some information about how this model is “placed” in the landscape of the other CMIP6 models. I know there are analyses presented showing the span of climate sensitivities in these models, but there may also be analyses of aerosol experiments, to give an indication of how sensitive the UKESM1 climate is to aerosols compared to other models?
- Table 1, line 3: It says “CMIP” – should it be “CMIP6”?
- L 175: “exerting global ERF” --> “exerting a global ERF”
- L 176: “The regional effect of aerosols” --> “The regional effect of aerosol changes in this period” (to make clear for the reader that you’re still talking about T1)
- L 178: The trend difference between EUS and ECC was really small – perhaps you should tone down the wording and say that “Particularly WEU shows a stronger aerosol ERF trend than ECC” or something like that?
- L 179: The sentence starting with “Since” is a bit strange and should be changed. “than during the PI period” --> “compared to the PI period”? And “has been excerted” --> “has been prominent”?
- L 192: What is mean with “a relatively similar value range was observed”? Please write out more specifically.
- L 218: Add a line shift after the table
- 4: In the caption, is d) supposed to say “all aerosols” and not just “aerosols”?
- L 235: While a closer look at the factors through which the aerosols influence the ERF is an excellent idea, I feel that this analysis brings little new. You have already shown the role of BC versus sulfate in the two regions, and shown that the ERF is mostly dominated by the RA, which again is mostly dominated by the cloud radiative effect. I think you need to state clearly what contributing factors you are going to look at in this final analysis.
- L 241: “seems to be highly correlated” – this should be supported by actual correlation numbers, e.g. by doing spatial correlations between the maps in Fig. 5.
- L 247: “due to the dominant effect of absorbing aerosols” – to convince the reader that this pattern stems from BC, it would be good to see maps of BC concentrations/emissions for this region..
- L 248: Please add actual correlation coefficients (and level of significance)
- L 249: “still seems to be the main contributor of IRFsw” ..but what else could it have been? What other factors have the potential to influence the IRFsw?
- L 265: The authors find an increase in stratus clouds in regions of strong BC changes, and explain this by increased stabilization. This should be supported by numbers or a figure showing the stabilization seen in UKESM1 – not just by referring to other papers..
- 6: Consider reversing the color scale so that blue means more clouds, which is more intuitive.
- L 289: “quantify” --> “quantified”
